



# Measurement report: Photochemical production and loss rates of formaldehyde and ozone across Europe

Clara M. Nussbaumer[1], John N. Crowley[1], Jan Schuladen[1], Jonathan Williams[1,2], Sascha Hafermann[1],
Andreas Reiffs[1], Raoul Axinte[1], Hartwig Harder[1], Cheryl Ernest[1,a], Anna Novelli[1,b], Katrin Sala[1],
Monica Martinez[1], Chinmay Mallik[1,c], Laura Tomsche[1,d], Christian Plass-Dülmer[3], Birger Bohn[4],
Jos Lelieveld[1,2], and Horst Fischer[1]

[1]Max Planck Institute for Chemistry, Department of Atmospheric Chemistry, 55128 Mainz, Germany
[2]Climate and Atmosphere Research Center, The Cyprus Institute, Nicosia, Cyprus
[3]German Meteorological Service, Meteorological Observatory Hohenpeissenberg (MOHp), 83282 Hohenpeissenberg,
Germany
[4]Institute of Energy and Climate Research, IEK-8: Troposphere, Forschungszentrum Jülich GmbH, 52428 Jülich, Germany
[a]now at: Department of Neurology, University Medical Center of the Johannes Gutenberg University Mainz, 55131 Mainz,
Germany
[b]now at: [4]
[c]now at: Department of Atmospheric Science, Central University of Rajasthan, Rajasthan 305817, India
[d]now at: German Aerospace Center, Institute of Atmospheric Physics, 82234 Wessling-Oberpfaffenhofen, Germany

**Correspondence:** Clara M. Nussbaumer (clara.nussbaumer@mpic.de)

**Abstract.** Various atmospheric sources and sinks regulate the abundance of tropospheric formaldehyde (HCHO) which is an important trace gas impacting the $HO_x$ ($\equiv HO_2 + OH$) budget and the concentration of ozone ($O_3$). In this study, we present the formation and destruction terms of ambient HCHO and $O_3$ calculated from in-situ observations of various atmospheric trace gases measured at three different sites across Europe during summer time. These include a coastal site in Cyprus in the scope of the Cyprus Photochemistry Experiment (CYPHEX) in 2014, a mountain site in Southern Germany as part of the Hohenpeißenberg Photochemistry Experiment (HOPE) in 2012 and a forested site in Finland where measurements were performed during the Hyytiälä United Measurements of Photochemistry and Particles (HUMPPA) campaign in 2010. We show that at all three sites formaldehyde production from the OH oxidation of methane ($CH_4$), acetaldehyde ($CH_3CHO$), isoprene ($C_5H_8$) and methanol ($CH_3OH$) can almost completely balance the observed loss via photolysis, OH oxidation and dry deposition. Ozone chemistry is clearly controlled by nitrogen oxides ($NO_x \equiv NO + NO_2$) that includes $O_3$ production from $NO_2$ photolysis and $O_3$ loss via the reaction with NO. Finally, we use the HCHO budget calculations to determine whether net ozone production is limited by the availability of VOCs (VOC limited regime) or $NO_x$ ($NO_x$ limited regime). At the mountain site in Germany $O_3$ production is VOC limited, whereas it is $NO_x$ limited at the coastal site in Cyprus. The forested site in Finland is in the transition regime.





## 1  Introduction

Formaldehyde (HCHO) is an important atmospheric trace gas, which provides insight into various photochemical processes taking place in the earth's atmosphere. It has both anthropogenic sources, such as industrial and vehicle emissions, and natural sources including for example biomass burning or VOC precursors, with natural sources dominating in remote locations (Luecken et al., 2018; Anderson et al., 2017; Stickler et al., 2006; Wittrock et al., 2006; Lowe and Schmidt, 1983). The majority

of these HCHO sources is secondary and due to its short lifetime, atmospheric transport of HCHO from primary (direct) emissions (e.g. biomass burning or industry) to remote locations can be mostly neglected (Fortems-Cheiney et al., 2012; Vigouroux et al., 2009; Anderson et al., 2017). Loss processes of HCHO include deposition, reaction with OH and photolysis yielding mainly $HO_2$, CO and $H_2$ (Anderson et al., 2017). HCHO production paths are more diverse and include oxidation processes of almost any volatile organic compound (VOC) including acetone ($CH_3COCH_3$), methane ($CH_4$), acetaldehyde ($CH_3CHO$),

methanol ($CH_3OH$), isoprene ($C_5H_8$), methyl hydroperoxide ($CH_3OOH$), ethene ($C_2H_4$) (these selected species are included in this study due to the availability of measurement data) and many more, the majority of which are initiated by the OH radical during the day. (Stickler et al., 2006; Wittrock et al., 2006). Net production processes of formaldehyde therefore influence the $HO_x$ ($HO_x \equiv OH + HO_2$) budget which in turn controls the atmospheric oxidizing capacity (Luecken et al., 2018). This includes the regulation of the atmospheric ozone ($O_3$) abundance, a trace gas with adverse health effects for humans, animals

and plants leading to cardiovascular and respiratory diseases and the loss of life expectancy (Nuvolone et al., 2018; Lippmann, 1989). It is therefore important to understand the processes influencing and contributing to HCHO and $O_3$ formation and loss processes in the earth's atmosphere (see also Figure 1 and 2 for an overview of the reactions considered in this study).

Previous studies have investigated the processes contributing to HCHO production from secondary sources. Palmer et al. (2003) identified isoprene, methane and methanol to be the main HCHO precursor over the United States of America contribut-

ing by over 80 % in the GEOS-CHEM model (Goddard Earth Observing System global 3-D model of tropospheric chemistry). Anderson et al. (2017) evaluated HCHO concentrations in the tropical western pacific and found methane and acetaldehyde to be the main precursors of HCHO based on box model simulations. Sumner et al. (2001) investigated the HCHO budget at a forest in Pellston, Michigan (U.S.A.) based on observations in the scope of PROPHET (Program for Research on Oxidants: Photochemistry, Emission and Transport) in 1998. They identified isoprene to be the main HCHO precursor with around 80 %.

Dienhart et al. (2021) investigated the relationship between OH reactivity and HCHO production rates during the shipborne campaign AQABA (Air Quality and Climate Change in the Arabian Basin) around the Arabian Peninsula in 2017 which they found to be highest in polluted areas, suggesting a high diversity of HCHO precursors. Kaiser et al. (2015) studied OH reactivities and HCHO concentrations in the Po Valley based on zeppelin measurements during the research campaign PEGASOS (Pan-European Gas-AeroSOls Climate Interaction Study) in 2012 in comparison with model simulations and attributed

discrepancies to possible primary HCHO emissions from agriculture.

Tropospheric ozone chemistry is dependent on the $O_3$ precursors $NO_x$ ($NO_x \equiv NO + NO_2$) and volatile organic compounds (VOC). Depending on the ambient concentrations of $NO_x$ and VOC, net ozone formation can either be $NO_x$ or VOC limited. A $NO_x$ limitation is usually dominant for low $NO_x$ concentrations in which increasing $NO_x$ leads to an increase in $O_3$ formation.



For high $NO_x$ concentrations, ozone formation is usually VOC limited and an increase in ambient $NO_x$ reduces $O_3$ formation
through loss of $HO_x$ as OH is converted to $HNO_3$ (Pusede et al., 2015; Nussbaumer and Cohen, 2020). Consequently, changes
in ambient $NO_x$ concentrations can either increase or decrease $O_3$ or - at a transition between both regimes - have only a weak
net effect on $O_3$ production. In urban environments, the chemical regime can be characterized using the weekend effect which
describes the ozone response to decreasing $NO_x$ emissions on weekends as reported by Pusede and Cohen (2012), Nussbaumer
and Cohen (2020), Pires (2012), Wang et al. (2014) and many more (Levitt and Chock, 1976; Seguel et al., 2012; Sadanaga
et al., 2011). Another measure for identifying the prevailing chemical regime is the ratio of HCHO and $NO_2$ which has been
determined via satellite measurements in various studies (Sillman, 1995; Jin et al., 2020; Martin et al., 2004; Duncan et al.,
2010; Jin and Holloway, 2015). Sillman (1995) initially suggested a threshold of 0.28 for the ratio $HCHO/NO_y$ ($NO_y$ is here
the sum of $NO_x$, $HNO_3$, peroxyacetylnitrates and alkyl nitrates), below which chemistry is VOC limited, based on model
simulations for Lake Michigan and the northeast corridor (United States of America). Martin et al. (2004) investigated the ratio
of the $HCHO/NO_2$ column based on satellite observations and found a threshold of 1. This is in agreement with findings from
Duncan et al. (2010) who suggested a threshold of 1 for a VOC limited regime, a threshold of 2 for a $NO_x$ limited regime and
a transition in between both using remote sensing. Schroeder et al. (2017) present in-situ measurements of HCHO and $NO_2$
for determining the dominant regime and points out that exact thresholds are geographically variable due to locally different
atmospheric composition and ambient conditions, such as VOC variety or humidity.

In this study, we evaluate the formaldehyde and ozone budget during the field experiment CYPHEX (Cyprus Photochemistry
Experiment) which took place in July 2014 at a coastal site in Cyprus (Ineia) based on in-situ trace gas observations of NO,
$NO_2$, $O_3$, OH, $HO_2$, $CH_4$, $CH_3OH$ (methanol), $C_5H_8$ (isoprene), $CH_3CHO$ (acetaldehyde), $CH_3COCH_3$ (acetone), $CH_3OOH$
(methyl hydroperoxide), $C_2H_4$ (ethene), $CH_3SCH_3$ (DMS) and HCHO (Derstroff et al., 2017; Meusel et al., 2016; Mallik et al.,
2018). We compare the results with two other field campaigns in central and northern Europe which are the Hohenpeißenberg
Photochemistry Experiment (HOPE 2012, Novelli et al. (2017)) at a mountain site in Germany and the Hyytiälä United Mea-
surements of Photochemistry and Particles (HUMPPA 2010, Williams et al. (2011)) at a boreal forest site in Finland. Only few
studies have evaluated the HCHO budget and mainly through model simulations. To our knowledge there is only one study by
Sumner et al. (2001) that has previously presented HCHO budget calculations from in-situ trace gas observations in the United
States of America in 1998. We are first to present HCHO budget calculations from in-situ measurements across Europe and
show that in all three locations HCHO production can be accounted for by the oxidation of methane, methanol, acetaldehyde
and isoprene.

## 2 Observations and methods

### 2.1 HCHO chemistry calculations

Figure 1 shows an overview of the main production and loss processes for formaldehyde which we consider in this study
and for which measurements were obtained during the field campaign CYPHEX (see Section 2.3.1 for further details). The
relationships we derive in this study are based on boundary layer conditions and we therefore assume no relevant intrusion



from higher altitudes. Acetone and methane can form methyl radicals ($CH_3$) through oxidation by OH or photolysis which are subsequently oxidized to methyl peroxy radicals ($CH_3O_2$) by molecular oxygen ($O_2$). Another pathway yielding $CH_3O_2$ is the OH oxidation of acetaldehyde which forms $CH_3C(O)$ in the first step which can then be oxidized to $CH_3C(O)O_2$ when $O_2$ is
present. $CH_3C(O)O_2$ yields $CH_3O_2$ through reaction with nitric oxide (NO) or the hydroperoxyl radical ($HO_2$) via $CH_3CO_2$. The $CH_3O_2$ yield from the reaction of $CH_3C(O)O_2$ with $HO_2$ is approximately 50 % ($k_c$). Other reactions pathways result in the formation of $CH_3C(O)OH$ ($k_a$) and $CH_3C(O)OOH$ ($k_b$) (IUPAC Task Group on Atmospheric Chemical Kinetic Data Evaluation, 2019). We calculate the overall fraction $\alpha_{CH_3CHO}$ of acetaldehyde oxidation that results in $CH_3O_2$ formation via Equation (1).

$$\alpha_{CH_3CHO} = \frac{k_{CH_3C(O)O_2+NO} \times [NO] + k_c \times [HO_2]}{k_{CH_3C(O)O_2+NO} \times [NO] + k_a \times [HO_2] + k_b \times [HO_2] + k_c \times [HO_2]} \qquad (1)$$

Additionally, methyl hydroperoxide ($CH_3OOH$) forms $CH_3O_2$ via OH oxidation (60 %) (or reacts directly to HCHO via photolysis or OH-initiated oxidation (40 %)) (IUPAC Task Group on Atmospheric Chemical Kinetic Data Evaluation, 2007). $CH_3O_2$ can then either react with $HO_2$ to form $CH_3OOH$ or yield HCHO through reaction with NO or OH via $CH_3O$ (Anderson et al., 2017; Stickler et al., 2006; Lowe and Schmidt, 1983; Fittschen et al., 2014). The importance of $CH_3O_2$ loss via OH in remote
locations has been recently shown in several studies and the reaction primarily yields HCHO (Lightfoot et al., 1992; Assaf et al., 2017, 2016; Fittschen et al., 2014; Yan et al., 2016). For simplification, we assume this yield to be 100 % which slightly increases the uncertainty of the calculation but is negligible given the small fraction of $CH_3O_2$ that reacts with OH ($< 10$ % for CYPHEX). In our study, we use the rate constant $k_{CH_3O_2+OH}$ recommended by the IUPAC Task Group on Atmospheric Chemical Kinetic Data Evaluation (2017). Table S1 of the Supplement gives an overview of all rate constants used in this study,
most of which were taken from IUPAC Task Group on Atmospheric Chemical Kinetic Data Evaluation (2021). The fraction of $CH_3O_2$ that forms HCHO ($\alpha_{CH_3O_2}$) is dependent on the ambient concentrations of $HO_2$, NO and OH and can be calculated via Equation (2). $CH_3O_2$ loss via self-reaction is negligibly small and therefore not included. The reaction with $NO_2$ forming $CH_3O_2NO_2$ can also be excluded due to its thermal instability in the boundary layer.

$$\alpha_{CH_3O_2} = \frac{k_{CH_3O_2+NO} \times [NO] + k_{CH_3O_2+OH} \times [OH]}{k_{CH_3O_2+NO} \times [NO] + k_{CH_3O_2+OH} \times [OH] + k_{CH_3O_2+HO_2} \times [HO_2]} \qquad (2)$$

Isoprene oxidation results in the formation of HCHO through the intermediate products methyl vinyl ketone and methacrolein as described by Wolfe et al. (2016). The HCHO yield from isoprene ($\alpha_{Isoprene}$) is dependent on the ambient NO concentration and varies between around 30 % for low $NO_x$ and 60 % for high $NO_x$ ($RO_2$ react primarily with NO) (Atkinson et al., 2006; Palmer et al., 2003; Sumner et al., 2001). We calculate the HCHO yield via Equation (3) and estimate $[HO_2] \approx [RO_2]$ as suggested by Sumner et al. (2001). Equating $HO_2$ to $RO_2$ increases the uncertainty of the analysis. However, the yield can only
vary between 34 % (in the absence of NO) and 57 % (when NO chemistry dominates the fate of the $RO_2$ formed) according to Equation (3) which was originally determined experimentally by Miyoshi et al. (1994). We discuss the effects of the threshold values in Section 3.1. $C_5H_8(OH)O_2$ is the peroxy radical resulting from isoprene oxidation and has six relevant isomers





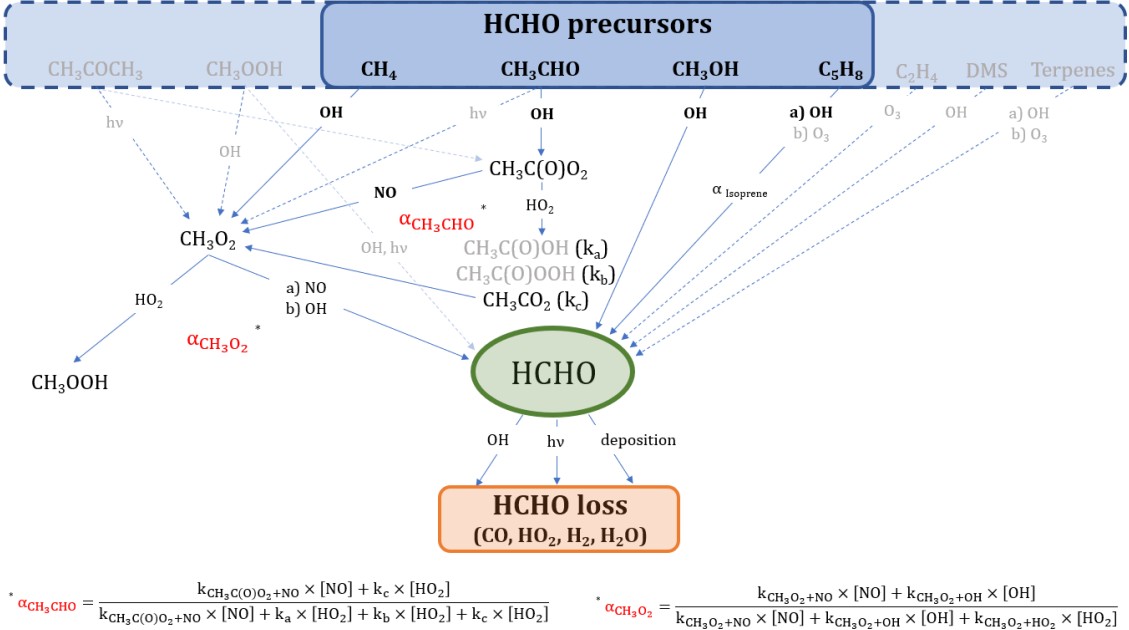

**Figure 1.** Overview of the chemical and photolytic reactions which lead to HCHO production and loss considering the trace gases measured during the field experiment CYPHEX. Black and bold font identifies the species which contribute mainly ($\sim$80 %) to HCHO formation according to the findings in this study. For a better overview, we have omitted intermediate steps with a 100 % yield which are instead described in the main text.

which can undergo multiple reactions yielding HCHO and many other products (Wennberg et al., 2018; Schwantes et al., 2020). Additionally, the formation of HCHO from isoprene does not occur instantaneously (as for example from methane or
other VOC), but is likely time-dependent. However, the consideration of this time-dependent formation as well as the detailed evaluation of the reactions paths from each peroxy radical isomer is beyond the scope of this study and we therefore adapt the methodology presented by Sumner et al. (2001) to estimate the HCHO production from isoprene.

$$\alpha_{Isoprene} = 0.34 + 0.23 \times \left( \frac{k_{C_5H_8(OH)O_2+NO} \times [NO]}{k_{C_5H_8(OH)O_2+NO} \times [NO] + k_{C_5H_8(OH)O_2+HO_2} \times [HO_2] + k_{C_5H_8(OH)O_2+RO_2} \times [RO_2]} \right)$$
(3)

Methanol reacts with OH yielding HCHO via $CH_2OH$ and $CH_3O$ and following oxidation by $O_2$ (Anderson et al., 2017;
Stickler et al., 2006). Ethene is a precursor to HCHO through OH oxidation or ozonolysis (Alam et al., 2011; Atkinson et al., 2006). A potential source of HCHO in marine environments is dimethyl sulfide (DMS) via OH oxidation (Ayers et al., 1997; Urbanski et al., 1997). Terpenes such as limonene or $\alpha$-/$\beta$-pinene emitted from plants can additionally be HCHO sources as described by Lee et al. (2006).





Reactions (R1) - (R3) present the chemical loss processes of HCHO through OH oxidation and two different photolysis
pathways. In addition, HCHO dry deposition - the uptake of HCHO by the earth's surface - plays a role in HCHO loss,
particularly during the night (Anderson et al., 2017; Possanzini et al., 2002; Sumner et al., 2001; Wesely and Hicks, 2000).
While HCHO loss can also occur via wet deposition as for example described by Seyfioglu et al. (2006) or via liquid-phase
reactions in cloud droplets as shown by Franco et al. (2021), this study investigates summertime campaigns without significant
precipitation.

$$HCHO + OH + O_2 \rightarrow CO + HO_2 + H_2O \tag{R1}$$

$$HCHO + h\nu \rightarrow CO + H_2 \tag{R2}$$

$$HCHO + h\nu + 2O_2 \rightarrow CO + 2HO_2 \tag{R3}$$

We will show in the scope of this work that reactions of methane, acetaldehyde, methanol and isoprene with OH al-
most completely account for HCHO production in the environments considered in this paper across Europe. We have high-
lighted these pathways in Figure 1 in black and bold font. Equations (4) and (5) show the calculation of the basic production
$P(HCHO)_{basic}$ (compared to the reactions shown in Figure 1) and the loss $L(HCHO)$ terms. $k$ values represent the rate co-
efficients, $j(HCHO)$ is the summed photolysis frequency for reactions (R2) and (R3), $v_d$ describes the dry deposition velocity
in $\text{cm}\,\text{s}^{-1}$ and $BLH$ is the boundary layer height in cm.

$$
\begin{aligned}
P(HCHO)_{basic} &= P_{CH_4+OH} + P_{CH_3CHO+OH} + P_{CH_3OH+OH} + P_{C_5H_8+OH} \\
&= [OH] \times ([CH_4] \times k_{CH_4+OH} \times \alpha_{CH_3O_2} + [CH_3CHO] \times k_{CH_3CHO+OH} \times \alpha_{CH_3O_2} \times \alpha_{CH_3CHO} \\
&+ [CH_3OH] \times k_{CH_3OH+OH} + [C_5H_8] \times k_{C_5H_8+OH} \times \alpha_{Isoprene})
\end{aligned}
\tag{4}
$$

$$
\begin{aligned}
L(HCHO) &= L_{HCHO+OH} + L_{HCHO+h\nu} + L_{deposition} \\
&= [HCHO] \times \left( [OH] \times k_{HCHO+OH} + j(HCHO) + \frac{v_d(HCHO)}{BLH} \right)
\end{aligned}
\tag{5}
$$

Changes in the HCHO concentration are represented by Equation (6) which includes production and loss from Equations (4) -
(5), a transport term T(HCHO) such as advection or entrainment (can be positive or negative) and a term for primary emissions
(Fischer et al., 2019).

$$\frac{d[HCHO]}{dt} = P(HCHO)_{basic} - L(HCHO) + T(HCHO) + P_{emission} \tag{6}$$





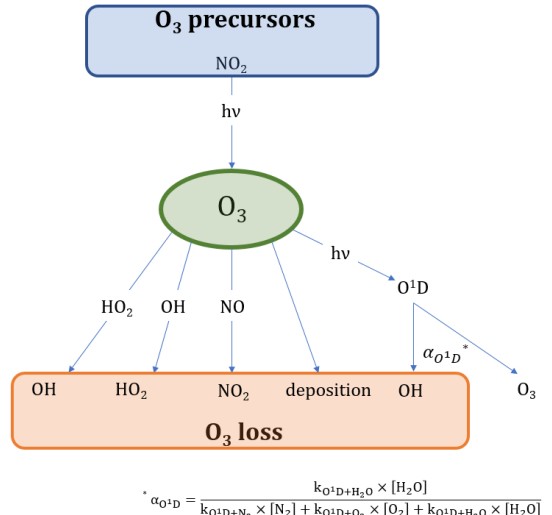

$$^*\alpha_{O^1D} = \frac{k_{O^1D+H_2O} \times [H_2O]}{k_{O^1D+N_2} \times [N_2] + k_{O^1D+O_2} \times [O_2] + k_{O^1D+H_2O} \times [H_2O]}$$

**Figure 2.** Overview of the chemical and photolytic reactions which lead to $O_3$ production and loss.

## 2.2 $O_3$ chemistry calculations

Figure 2 presents the main processes contributing to $O_3$ formation and loss. The only significant chemical source of tropospheric $O_3$ is the photolysis of nitrogen dioxide ($NO_2$) which is converted to NO and $O(^3P)$ under the influence of sunlight. The reaction of $O(^3P)$ with molecular oxygen ($O_2$) subsequently yields $O_3$ (Jacob, 1999). $NO_2$ in turn is generated from the oxidation of NO by ozone or peroxy radicals ($HO_2$, $RO_2$) (Pusede et al., 2015).

Ozone loss processes include the reaction with NO forming $NO_2$, conversion with OH or $HO_2$ to $HO_2$ and OH, respectively, deposition processes and photolysis. $O_3$ photolysis yields $O(^1D)$ which is deactivated to $O(^3P)$ (and then $O_3$) via collision with nitrogen ($N_2$) and oxygen ($O_2$). $O_3$ loss occurs when $O(^1D)$ reacts with water to form OH. The fraction of this reaction is presented by $\alpha_{O^1D}$ as shown in Equation (7) (Bozem et al., 2017). The reaction of $O_3$ and $NO_2$ forming $NO_3$ could potentially yield a net loss of $O_3$ when being photolyzed back to $NO_x$. However, only around 10 % of the $NO_3$ photolysis leads to NO formation (which results in a $O_3$ net loss) (Stockwell and Calvert, 1983). Additionally, the reaction is likely negligible during the day as $O_3$ reacts much more rapidly with NO.

$$\alpha_{O^1D} = \frac{k_{O^1D+H_2O} \times [H_2O]}{k_{O^1D+N_2} \times [N_2] + k_{O^1D+O_2} \times [O_2] + k_{O^1D+H_2O} \times [H_2O]} \tag{7}$$

Equations (8) and (9) present the calculations for $O_3$ production and loss.

$$P(O_3) = P_{NO_2+h\nu} = [NO_2] \times j(NO_2) \tag{8}$$





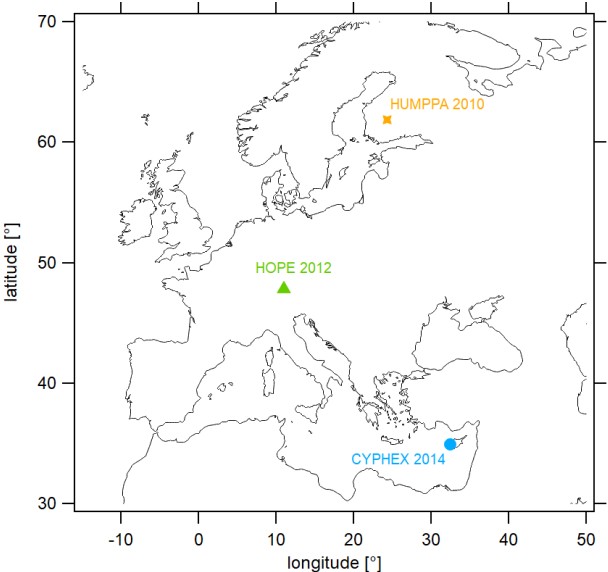

**Figure 3.** Geographic locations of the measurement sites included in this analysis. CYPHEX: 34.96 °N, 32.38 °E, 650 m a.s.l. (above sea leavel), UTC + 128 min; HOPE: 47.80 °N, 11.01 °E, 980 m a.s.l., UTC + 44 min; HUMPPA: 61.85°N, 24.28 °E, 181 m a.s.l., UTC + 96 min.

$$L(O_3) = L_{O_3+NO} + L_{O_3+OH} + L_{O_3+HO_2} + L_{O_3+h\nu} + L_{deposition}$$

$$= [O_3] \times \left( [NO] \times k_{O_3+NO} + [OH] \times k_{O_3+OH} + [HO_2] \times k_{O_3+HO_2} + j(O^1D) \times \alpha_{O^1D} + \frac{v_d(O_3)}{BLH} \right) \qquad (9)$$

As for HCHO, net changes in $O_3$ are represented by production, loss and a transport term (either positive or negative) as shown in Equation (10).

$$\frac{d[O_3]}{dt} = P(O_3) - L(O_3) + T(O_3) \qquad (10)$$

### 2.3 Field experiments

We have analyzed trace gases and further measurement parameters in regard to the HCHO and $O_3$ budget at three different measurement sites across Europe which are located in Cyprus (CYPHEX campaign 2014), Southern Germany (HOPE campaign 2012) and Finland (HUMPPA campaign 2010). Their geographic locations are shown in Figure 3. We provide details on each campaign in the following sections. Please note that all times are in UTC (coordinated universal time). The time difference between 12:00 UTC and local noon are UTC + 128 min for Cyprus, UTC + 44 min for Germany and UTC + 96 min for Finland (Fischer et al., 2019).





### 2.3.1 CYPHEX campaign 2014

The Cyprus Photochemistry Experiment (CYPHEX) took place in Ineia, Cyprus in July and August 2014. The measurement site was situated on a hilltop 650 m above sea level (34.96 °N, 32.38 °E, UTC + 128 min) in a remote location with low population in the surrounding areas. The distance to the coastline of the Mediterranean Sea was approximately 10 km in the North and in the West. A detailed description of the measurement site can be found in Derstroff et al. (2017), Meusel et al. (2016) and Mallik et al. (2018). In this study, we consider the campaign days for which the trace gas measurements were available simultaneously which is the time period 22 - 31 July 2014. NO and $NO_2$ were measured via photolytic chemiluminescence (detector from ECO Physics CLD 790 SR, Dürnten Switzerland, photolytic converter from Droplet Measurement Technologies, Boulder, USA) with a total uncertainty of 20 and 30 % and a detection limit of 5 and 20 $ppt_v$, respectively (Hosaynali Beygi et al., 2011; Tadic et al., 2020). $O_3$ was measured via UV photometry ($O_3$-Analyzer model 49, Thermo Environment Instruments, USA) with a detection limit of 2 $ppb_v$ and a total uncertainty of 5 %. OH and $HO_2$ were measured via laser-induced fluorescence spectroscopy with the custom-built HORUS (HydrOxyl Radical measurement Unit based on fluorescence Spectroscopy) instrument (accuracy 28.5 and 36 % & detection limit $1 \times 10^6$ molec cm$^{-3}$ and 0.8 $ppt_v$, respectively) (Marno et al., 2020; Novelli et al., 2014). HCHO was measured via the method of Hantzsch with a commercial instrument (Aero-Laser model AL 4021, Garmisch Partenkirchen, Germany) with a detection limit of 38 $ppt_v$ and a total uncertainty of 16 % (Kormann et al., 2003). $C_2H_4$ and $CH_4$ were determined via gas chromatography flame ionization detection (GC 5000 VOC, AMA instruments, Ulm, Germany). $CH_4$ measurements had a detection limit of 20 $ppb_v$ and a total uncertainty of 2 % and $C_2H_4$ measurements had a detection limit of 1 - 8 $ppt_v$ and a total uncertainty of 10 % (Sobanski et al., 2016; Mallik et al., 2018). Isoprene was measured via gas chromatography mass spectrometry (MSD 5973, Agilent Technologies GmbH, Böblingen, Deutschland) with a detection limit of 1 $ppt_v$ and a total uncertainty of 14.5 % (Derstroff et al., 2017). Oxygenated VOC (OVOC) were measured via proton-transfer-reaction time-of-flight mass spectrometry (PTR-TOF-MS) (Ionicon Analytik GmbH, Innsbruck, Austria) ($CH_3OH$: 242 $ppt_v$ limit of detection (LOD), 37 % total uncertainty (TU); $CH_3COCH_3$: 97 $ppt_v$ LOD, 10 % TU; $CH_3CHO$: 85 $ppt_v$ LOD, 22 % TU; DMS: 18 $ppt_v$ LOD, 12 % TU; the TU is 4 - 7 % higher for a relative humidity below 25 %) (Veres et al., 2013; Graus et al., 2010). $CH_3OOH$ was measured via high-performance liquid chromatography with a detection limit of 25 $ppt_v$ and a total uncertainty of 9 %. Photolysis frequencies were determined with a single monochromator spectral radiometer (Meterologie Consult GmbH, Königstein, Germany) with a total uncertainty of around 10 %. The photolysis frequencies for acetaldehyde j($CH_3CHO$) and formaldehyde j(HCHO) were determined via parameterizations by the help of j($NO_2$) and j($O^1D$) using the latest IUPAC quantum yield data (2013) for which we estimate an uncertainty of around 20 %. An overview of all measured trace gases including the measurement uncertainty and the time resolution of the data used in this study can be found in Table S2 of the Supplement.

### 2.3.2 HOPE campaign 2012

The Hoßenpeißenberg Photochemistry Experiment (HOPE) took place from June to September 2012 in Hohenpeißenberg, Germany at the Global Atmospheric Watch Meteorological Observatory (47.80 °N, 11.01 °E, UTC + 44 min). The measure-





ment location was situated on a hilltop 980 m above sea level in a remote and vegetated area. More details on the campaign

and the site location can be found in Novelli et al. (2017). $O_3$, HCHO, OH and $HO_2$ were measured with the same methods as described for the CYPHEX campaign in Section 2.3.1. NO and $NO_2$ were measured via (photolysis) chemiluminescence by the German Weather Service with an estimated uncertainty of 10 %. $CH_4$ was measured via GC-FID. Isoprene and OVOC were determined via a custom built GC (Agilent Technologies) FID/MS system(Novelli et al., 2017; Werner et al., 2013). j($NO_2$) and j($O^1D$) were measured with filter radiometers (Meteorologie Consult GmbH, Königstein, Germany) with an uncertainty

of 10 % (Bohn et al., 2008). j(HCHO) was determined via parameterization using the latest IUPAC quantum yield data (2013) with an uncertainty of 20 %. BLH measurements were not performed and instead adopted from Fischer et al. (2019) which are 1500 m for daytime (j($NO_2$) > $10^{-3}$ s$^{-1}$) and 200 m for nighttime (j($NO_2$) < $10^{-3}$ s$^{-1}$). We determined the deposition velocity $v_d$ from the HCHO nighttime loss as previously performed by Fischer et al. (2019) for $H_2O_2$. The loss rate coefficient $k_d$ was determined from the [HCHO] decrease from 21:00 - 01:30 UTC divided by the average HCHO concentration [HCHO]$_{av}$ in

this time interval. Multiplication with the nighttime boundary layer height (BLH) (200 m) then yielded the deposition velocity according to Equation (11). The HCHO loss via deposition during nighttime is independent of the BLH (see Equation (5) and (11)).

$$v_d = \frac{k_{d,night} \times BLH_{night}}{x} = \frac{\frac{d[HCHO]}{dt}}{[HCHO]_{av} \times x} \times BLH_{night} \qquad (11)$$

Please note that $v_d$ derived this way represents a lower limit to the nighttime loss rate as HCHO could be formed from $NO_3$ and

$O_3$ chemistry (for example from ozone and isoprene) at night (Crowley et al., 2018). The factor $x$ considers the inconsistent mixing of the boundary layer at night for which $x$ was 2, assuming a linear gradient between the top and the bottom of the boundary layer. During the day, $x$ equaled 1 (Shepson et al., 1992; Fischer et al., 2019). As we determined the loss rate from the nighttime decrease in HCHO, the daytime deposition velocity was twice the nighttime deposition velocity according to Equation (11) which gives $v_d$(day) = 0.40 cm s$^{-1}$ and $v_d$(night) = 0.20 cm s$^{-1}$. Literature values for daytime deposition veloc-

ities range between 0.36 and 1.5 cm s$^{-1}$ and for nighttime between 0.18 and 0.65 cm s$^{-1}$ (Sumner et al., 2001; Stickler et al., 2007; DiGangi et al., 2011; Ayers et al., 1997). We therefore consider our calculation to yield reasonable estimates.

We have performed this calculation for nine nights. Figure 4 exemplarily shows the nighttime loss of HCHO for one night during the HOPE campaign. Red data points represent the HCHO mixing ratios, black color highlights the data points which we included in our analysis (loss between 21:00 and 01:30 UTC) and the cyan line is the linear fit of these data points. An overview

of the nights thus analyzed can be found in Figure S1b of the Supplement. The uncertainty of each calculation results from the single uncertainties of $\frac{d[HCHO]}{dt}$, [HCHO]$_{av}$ and the nighttime BLH. The uncertainty of $\frac{d[HCHO]}{dt}$ is composed of the HCHO measurement uncertainty (16 %) and the uncertainty of the fit (30 % upper limit) with the latter dominating. The uncertainty of [HCHO]$_{av}$ is based on the HCHO measurement uncertainty and the HCHO averaging (20 % upper limit). Again, the uncertainty of the averaging prevails over the measurement uncertainty. The uncertainty of the BLH is 20 %. Gaussian error

propagation gives an overall uncertainty of $\sqrt{30\%^2 + 20\%^2 + 20\%^2} = 41\%$ from the calculation. The uncertainty from the atmospheric variability represented by the 1 $\sigma$ standard deviation of the mean over the considered nine nights is 58 % which

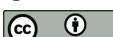



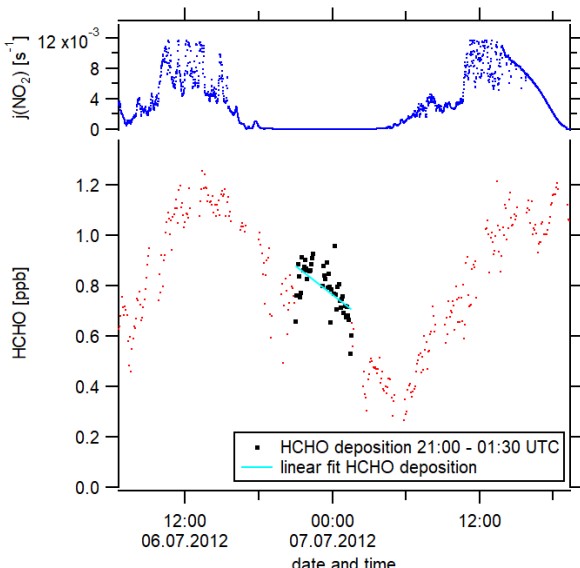

**Figure 4.** The determination of the deposition velocity is based on the HCHO nighttime loss - here exemplarily shown for one night during the campaign HOPE (2012) in Hohenpeißenberg, Southern Germany.

exceeds the uncertainty from the calculation. We therefore estimate the uncertainty of $v_d$ to be 58 %. Please note that in our uncertainty analysis we consider the arising statistical errors but not the systematic errors which are not quantifiable but could potentially increase the overall uncertainty. An overview of all uncertainties and the time resolution of the measured trace gases

can be found in Table S2 of the Supplement. Absolute $H_2O$ concentrations were estimated from the measured relative humidity via the Magnus formula over water. Please note that some trace gases were not available simultaneously or only for a short overlap. We therefore present the averaged diurnal profiles.

### 2.3.3 HUMPPA campaign 2010

The Hyytiälä United Measurements of Photochemistry and Particles (HUMPPA) campaign took place in July and August 2010

at SMEAR II (Station for Measuring Ecosystem-Atmosphere Relation) in Hyytiälä, Finland (61.85°N, 24.28 °E, UTC + 96 min, 181 m above sea level). The site is located in a remote area in a boreal forest. A detailed description of the campaign can be found in Williams et al. (2011). The measurement methods for $NO_x$, $O_3$, HCHO and $CH_4$ were the same as presented for the CYPHEX campaign. $CH_3OH$ was measured via ColdTrap PTR-MS with a detection limit of around 50 $ppt_v$. Acetaldehyde was measured via PTR-MS with a detection limit of 50 $ppt_v$ (Williams et al., 2011). Isoprene was measured via GC (GC 6890A,

Agilent Technologies) coupled to a Mass Selective Detector (MSD 5973 inert, Agilent Technologies) with an uncertainty of around 15 %. Highly constrained box-model simulations which were in fair agreement with experimental data where available for OH and $HO_2$ as described by Crowley et al. (2018) with an uncertainty in the order of 30 - 40 %. $j(NO_2)$ and $j(O^1D)$ were measured with filter radiometers (Meteorologie Consult GmbH, Königstein, Germany) with an uncertainty of around



10 %; j(HCHO) was determined via parameterization for which we used the 2013 IUPAC quantum yield data (in analogy
to CYPHEX and HOPE) (Bohn et al., 2008). $H_2O$ was measured with an infrared light absorption analyser (URAS 4 H2O,
Hartmann & Braun, Frankfurt am Main, Germany). The boundary layer height was measured by radio soundings as presented
by Ouwersloot et al. (2012) ranging from 200 m during nighttime (here j($NO_2$) < $10^{-3}$ s$^{-1}$) to 1500 m during daytime (here
j($NO_2$) < $10^{-3}$ s$^{-1}$) (Fischer et al., 2019). The deposition velocity was determined in analogy to the HOPE campaign based on
the HCHO loss from 00:00 - 04:30 UTC on the basis of 15 nights and was 0.72 cm s$^{-1}$ during the day and 0.36 cm s$^{-1}$ during the
night. The uncertainty of the calculation results from the single uncertainties of $\frac{d[HCHO]}{dt}$ (16 %), $[HCHO]_{av}$ (33 %) and the
nighttime BLH (20 %). Gaussian error propagation gives an overall uncertainty of 42 % from the calculation. The atmospheric
variability equals 43 %. The determined uncertainties are almost the same and we therefore estimate the total uncertainty of
the deposition velocity to be 43 %. An overview of all considered nights and the according HCHO loss is presented in Figure
S1c of the Supplement. The deposition velocity of ozone was adopted from Rannik et al. (2012) for the time period W (week)
25 - 34 of the year and a relative humidity below 70 % which is 0.491 cm s$^{-1}$ for daytime and 0.069 cm s$^{-1}$ for nighttime. A
modeling study by Emmerichs et al. (2021) presents average values in the same order of magnitude ($\sim$ 0.2 - 0.3 cm s$^{-1}$ for
July and August). Again, uncertainties and time resolution are shown in Table S2 of the Supplement. Similar to the HOPE
campaign, some trace gases were not available simultaneously and we therefore present the averaged diurnal profiles.

## 3 Results and Discussion

### 3.1 Net HCHO production during CYPHEX 2014

HCHO concentrations during the CYPHEX campaign ranged between 0.3 and 1.9 ppb$_v$ (1.1 $\pm$ 0.4 ppb$_v$ on average) with a
maximum in the diel cycle during morning hours (04:00 UTC) and a minimum in the afternoon (15:00 UTC). The temporal
development of HCHO concentrations during the campaign and the respective j($NO_2$) values (to illustrate the daily cycle) are
shown in Figure S1a of the Supplement. Figure S2a of the Supplement presents the diurnal average of HCHO including its
rate of change dHCHO/dt. The uncertainty is dominated by the atmospheric variability which is on average 27 % for daytime
HCHO.

   Figure 5 shows the time series of (a) the HCHO production terms, (b) the HCHO loss terms and (c) the net HCHO production
during the research campaign CYPHEX in Cyprus in July 2014. We calculated the HCHO production terms for all measured
gas-phase precursors shown in Figure 1. The total measurement uncertainties were determined via gaussian error propagation
and were 28 % for HCHO production (31 % for only considering the OH oxidation of methane, methanol, isoprene and ac-
etaldehyde) and 26 % for HCHO loss. Table S3 of the Supplement provides an overview of all calculated uncertainties. We
present two example step-by-step calculations via gaussian error propagation in Equations (S1)-(S7) of the Supplement. All
other calculations were made accordingly. We have neglected the uncertainties for the HCHO yield from isoprene, $\alpha_{CH_3O_2}$ and
$\alpha_{CH_3CHO}$ and instead present a sensitivity study for these parameters later in this section. Dark green colors show the overall
HCHO production as sum of all single production terms. All production terms show a diurnal cycle which follows the course
of the photolysis frequency j($NO_2$) which we show in Figure 5d. Photolysis reactions do not take place after sunset and oxi-





(a) HCHO production

(b) HCHO loss

(c) net HCHO production

(d) NO$_2$ photolysis frequency j(NO$_2$) as illustration of the diurnal cycle

**Figure 5.** Temporal development of HCHO production and loss terms from July 22 to July 31, 2014 during the research campaign CYPHEX in Cyprus.

dation reactions are dependent on the abundance of OH radicals which is low during nighttime. Therefore, HCHO production approaches zero during nighttime. Due to data gaps a full diurnal profile was only available for July 23, July 28 and July 30





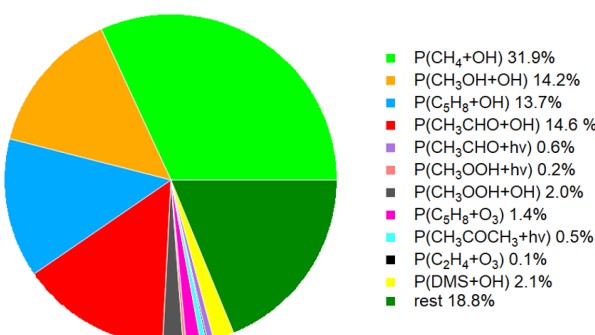

**Figure 6.** Chemical production terms of HCHO during CYPHEX.

where the overall HCHO production reached maxima between 0.6 and 0.8 $\mathrm{ppb}_v\,\mathrm{h}^{-1}$. The reactions of methane, methanol, iso-
290    prene and acetaldehyde with OH dominated the HCHO production processes. This is additionally illustrated in Figure 6 which
presents the average share of each production term based on a balance to the overall loss rate. HCHO production was therefore
dominated by the reaction of methane and OH (almost a third), followed by the oxidation of acetaldeyhde contributing around
15 % and the OH oxidations of isoprene and methanol, both contributing around 14 %. These four species together represented
75 % of the overall HCHO production required to balance the sinks. The production through OH oxidation of methyl hydroper-
295    oxide and dimethyl sulfide and through the reaction of isoprene and ozone contributed by around 1 - 2 % each. The remaining
species each yielded less than 1 % of the overall HCHO. Less than 20 % was unaccounted for ("rest" in Figure 6). This part also
includes HCHO production from terpenes via oxidation through OH or $O_3$. The yields from these reactions vary greatly in the
literature. Considering the yields from OH oxidation suggested by Lee et al. (2006) from laboratory investigations, limonene,
$\beta$- and $\alpha$-pinene would account for 3, 2 and 1 % of the overall HCHO production, respectively. The isoprene yield is limited to
300    a value between 34 and 57 %. The lower limit would give a HCHO production from isoprene of 11 % and the upper limit would
yield 19 %. The value for $\alpha_{CH_3O_2}$ can theoretically be between 0 and 100 % but is likely situated at the upper end due to the
availability of NO. A 20 % decrease in $\alpha_{CH_3O_2}$ would give a HCHO production from $CH_3O_2$ of 38 % (compare to 47 % for the
calculated $\alpha_{CH_3O_2}$). A 20 % increase would yield 57 % on average and decrease the "rest" to less than 10 %. For $\alpha_{CH_3CHO}$, a
20 % decrease and increase would give a HCHO yield from acetaldehyde of 12 % and 18 %, respectively, assuming a constant
$\alpha_{CH_3O_2}$. Please note that the uncertainty of the absolute values used to create this pie chart is dominated by the atmospheric
and diurnal variability of the single terms and is in the order of 100 %.

     HCHO loss was determined from the reaction with OH and photolysis. Red colors in Figure 5b show the overall calculated
HCHO loss. Similarly to HCHO production, HCHO loss via OH and photolysis only plays a role during daytime. At night, dry
deposition is the only HCHO loss mechanism and the deposition velocity $v_d$ can be determined from the nighttime decrease of
ambient HCHO concentrations as shown by Sumner et al. (2001). However, we often observed increasing HCHO concentra-
tions during nighttime, particularly before sunrise. A possible explanation could be local emissions due to traffic as personnel
arrived at the site during early morning hours since we observe a similar increase in $NO_2$ (possibly from NO titration with $O_3$)





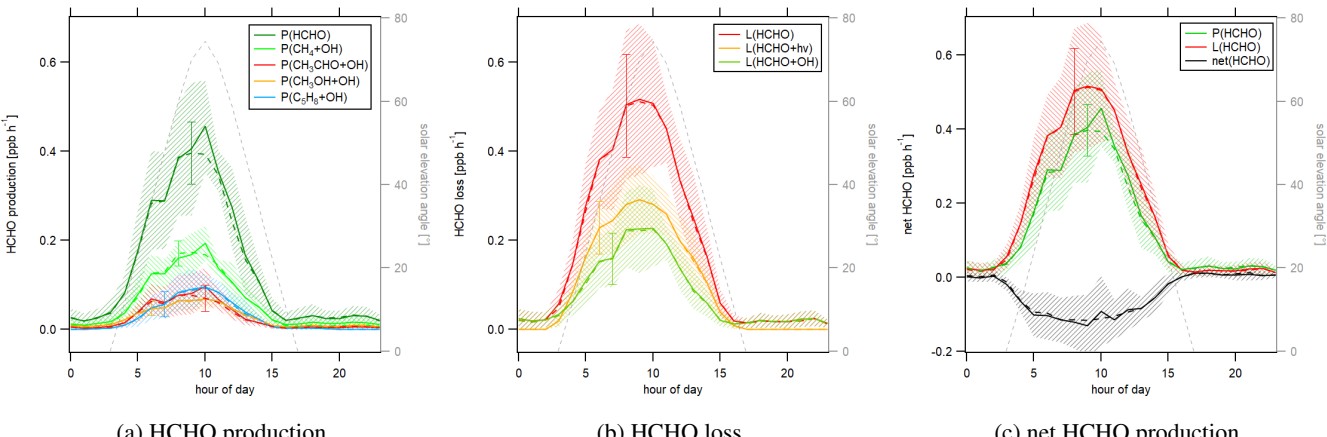

| (a) HCHO production | (b) HCHO loss | (c) net HCHO production |

**Figure 7.** Diurnal profiles of HCHO production P(HCHO) and loss L(HCHO) terms during the CYPHEX campaign 2014. Solid lines show the hourly averaged HCHO production and loss terms based on the point-by-point calculation of simultaneous parameter measurements. The uncertainty from the atmospheric variability is represented by the $1\sigma$ error shades. For the dashed lines, all measurement parameters were averaged first, followed by the calculation of hourly HCHO production and loss terms. The uncertainty is based on the atmospheric variability of all parameters and exemplarily shown by the error bar for one point for each production and loss term. The gray dashed line represents the diurnal solar elevation angle at the measurement site as determined by help of the NOAA Solar Calculator (2021).

and CO concentrations. The role of deposition processes at elevated altitudes is not yet fully understood but is likely influenced by flows along mountain slopes as well as horizontal advection on hilltops. For hilltops, incoming air through advection likely
originated from areas without deposition (too high above the ground) (Derstroff et al., 2017). As we do not observe net loss of HCHO at night, we estimate dry deposition to be negligible during the CYPHEX campaign. Figure 5c shows the overall HCHO production and loss rates as well as the difference between both, which we refer to as net HCHO production, in black color. Production and loss showed a good balance with values of $\pm 0.2\,\mathrm{ppb}_v\,\mathrm{h}^{-1}$. On most days, HCHO loss prevailed over HCHO production based on measured precursors.

In order to better account for diurnal changes we investigated the daily cycle of HCHO production and loss rates based on hourly averages over all measurement days. Figure 7 shows the diurnal profiles of (a) the main HCHO production terms as determined above, (b) the HCHO loss terms and (c) net HCHO production. The solid lines represent the hourly averages of the point-by-point calculation of the production and loss terms. The uncertainty is composed of the measurement uncertainty and the atmospheric variability, the latter demonstrated by the $1\sigma$ error shades. For HCHO production, the measurement uncertainty
was 31 % and the daytime atmospheric variability was 40 % (61 % for day and night). For HCHO loss, the uncertainties were 26 % and 34 % (57 %), respectively. In both cases, the overall uncertainty was dominated by the atmospheric variability. The dashed lines show the hourly production and loss terms calculated from the hourly averaged trace gas concentrations. The uncertainty from the atmospheric variability was 22 % (34 %) for HCHO production and 25 % (52 %) for HCHO loss, exemplarily indicated by an error bar for one point for each production and loss term in Figure 7. Both methods have advantages
and disadvantages. The point-by-point calculation allows for a simultaneous consideration of all measurement parameters.



Potential atmospheric changes or incidents such as winds, precipitation or rare primary emissions events are reflected by all parameters as they are monitored at the same time. On the other hand, this method reduces the amount of data points used as only one single missing species prevents the calculation of the overall term. In order to increase the number of results we have interpolated the values for the isoprene yield (data coverage 81 %), $\alpha_{CH_3CHO}$ and $\alpha_{CH_3O_2}$ (data coverage 64 %). Calculating

production and loss terms from the hourly averaged trace gas concentrations allows for the inclusion of all measured parameters but could potentially increase the uncertainty of the estimate depending on the duration and overlap of the single measurements. However in this case, the data were limited to one week in July with mainly simultaneous measurement of all parameters and we therefore do not expect large uncertainties. The similarity of the results as shown in Figure 7 indicates that both methods provide reasonable results.

For HCHO production, daily maxima of around $0.4 - 0.45\,\mathrm{ppb}_v\,\mathrm{h}^{-1}$ were reached between 9:00 and 10:00 UTC which was coincident with the maximum of the photolysis frequency j(NO$_2$). HCHO production from methane dominated the overall production term with a maximum of close to $0.2\,\mathrm{ppb}_v\,\mathrm{h}^{-1}$, followed by acetaldehyde, isoprene and methanol. HCHO loss peaked around 9:00 UTC with a value of $0.5\,\mathrm{ppb}_v\,\mathrm{h}^{-1}$ with around 55 % contribution from photolysis and 45 % from OH oxidation. Figure 7c shows that the calculated production term for HCHO can almost completely balance the loss which, assuming that

the major loss processes are well constrained, leads to the conclusion that HCHO production is well approximated by OH oxidation of methane, acetaldehyde, isoprene and methanol. This is in line with the HCHO rate of change presented in Figure S2a of the Supplement which fluctuated evenly around zero. Therefore, transport processes and primary emissions during CYPHEX can likely be excluded.

### 3.2 Net O$_3$ production during CYPHEX 2014

Ozone varied between 46 and $104\,\mathrm{ppb}_v$ during the CYPHEX campaign ($70 \pm 13\,\mathrm{ppb}_v$ on average) with a diel maximum at 04:00 UTC and a minimum at 15:00 UTC. The time series of O$_3$ concentrations are presented in Figure S3a of the Supplement. The diel mean including the rate of change dO$_3$/dt can be found in Figure S4a of the Supplement. The uncertainty is dominated by the atmospheric variability which is on average 16 % (1 $\sigma$) for O$_3$.

Figure 8 shows the diurnal cycle of the production and loss rates for ozone, analogous to Figure 7. Solid lines show the

hourly averaged point-by-point calculations of the O$_3$ production and loss terms. The 1 $\sigma$ error shades present the atmospheric variability which was 68 % for O$_3$ production and 37 % for O$_3$ daytime loss (44 % for day and night) The measurement uncertainties were 32 % and 16 %, respectively, and the overall uncertainty was therefore dominated by the atmospheric variability. Dashed lines show the O$_3$ production and loss terms based on the hourly averaged measurement parameters. The uncertainty resulting from the averaging of the individual parameters is exemplarily shown by the error bar for one point and is similar

to the atmospheric variability obtained by the point-by-point method (66 % for P(O$_3$) and 34 % (49 %) for L(O$_3$)). Figure 8a shows O$_3$ production represented by the photolysis of NO$_2$. Figure 8b shows the O$_3$ loss terms. O$_3$ loss through reaction with NO was dominant, followed by photolysis. The loss via OH and HO$_2$ was negligibly small. O$_3$ net production during the CYPHEX campaign was therefore clearly dominated by nitrogen oxides chemistry. Figure 8c shows net O$_3$ production. O$_3$ production and loss were similar throughout the day with peak values of $4 - 7\,\mathrm{ppb}_v\,\mathrm{h}^{-1}$ between 5:00 and 10:00 UTC. Net



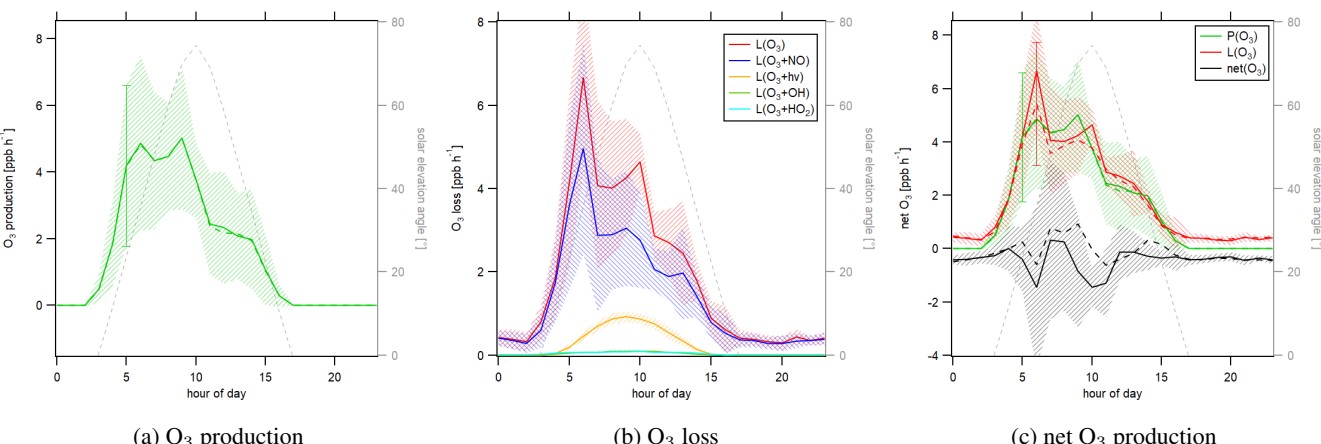

| (a) O$_3$ production | (b) O$_3$ loss | (c) net O$_3$ production |

**Figure 8.** Diurnal profiles of ozone production P(O$_3$) and loss L(O$_3$) terms during the CYPHEX campaign 2014. Solid lines show the hourly averaged O$_3$ production and loss terms based on the point-by-point calculation of simultaneous parameter measurements. The uncertainty from the atmospheric variability is represented by the 1 $\sigma$ error shades. For the dashed lines, all measurement parameters were averaged first, followed by the calculation of hourly O$_3$ production and loss terms. The uncertainty is mostly dominated by the atmospheric variability of all parameters and exemplarily shown by the error bar for one point for each production and loss term. Please find the temporal development of production and loss terms throughout the campaign in Figure S5 of the Supplement.

O$_3$ production was between -1 and 1 ppb$_v$ h$^{-1}$. Please note that we have excluded the NO$_2$ data on July 24 between 13:15 and 16:15 UTC due to a singular high concentration event. The large spike in NO$_2$ (O$_3$ titration of NO) is likely the result of sampling air impacted by the exhaust from a diesel generator which provided on site power and was located about 200 m from the containers housing the instruments. We show O$_3$ net production including all NO$_2$ data points in Figure S6a of the Supplement. O$_3$ production is directly proportional to the NO$_2$ concentration according to Equation (8) explaining the large

afternoon (14:00 UTC) production peak. We show the diel profile of NO$_2$ concentrations with and without the afternoon peak in Figure S6b of the Supplement. Production and loss terms for O$_3$ were balanced and the O$_3$ rate of change fluctuated evenly around zero (Figure S4a of the Supplement) suggesting that the diel variability was likely not impacted by transport processes.

## 3.3   Comparison with HOPE 2012 and HUMPPA 2010

### 3.3.1   Net HCHO production

We have shown in Sections 3.1 and 3.2 that the hourly averaging of measurement parameters and subsequent calculation of production and loss terms yielded reliable results during the CYPHEX campaign. For HUMPPA and HOPE, we have pursued this approach regarding the calculations of HCHO and O$_3$ production and loss terms, as the simultaneous data availability for all measurement parameters was too low for a point-by-point analysis.

     HCHO mixing ratios during the HOPE campaign varied between 0.1 and 3.2 ppb$_v$ (1.1 ± 0.5 ppb$_v$ on average) with a maxi-
mum in the diel cycle at 15:00 UTC and a minimum at 06:00 UTC. The average mixing ratio was similar to the average mixing

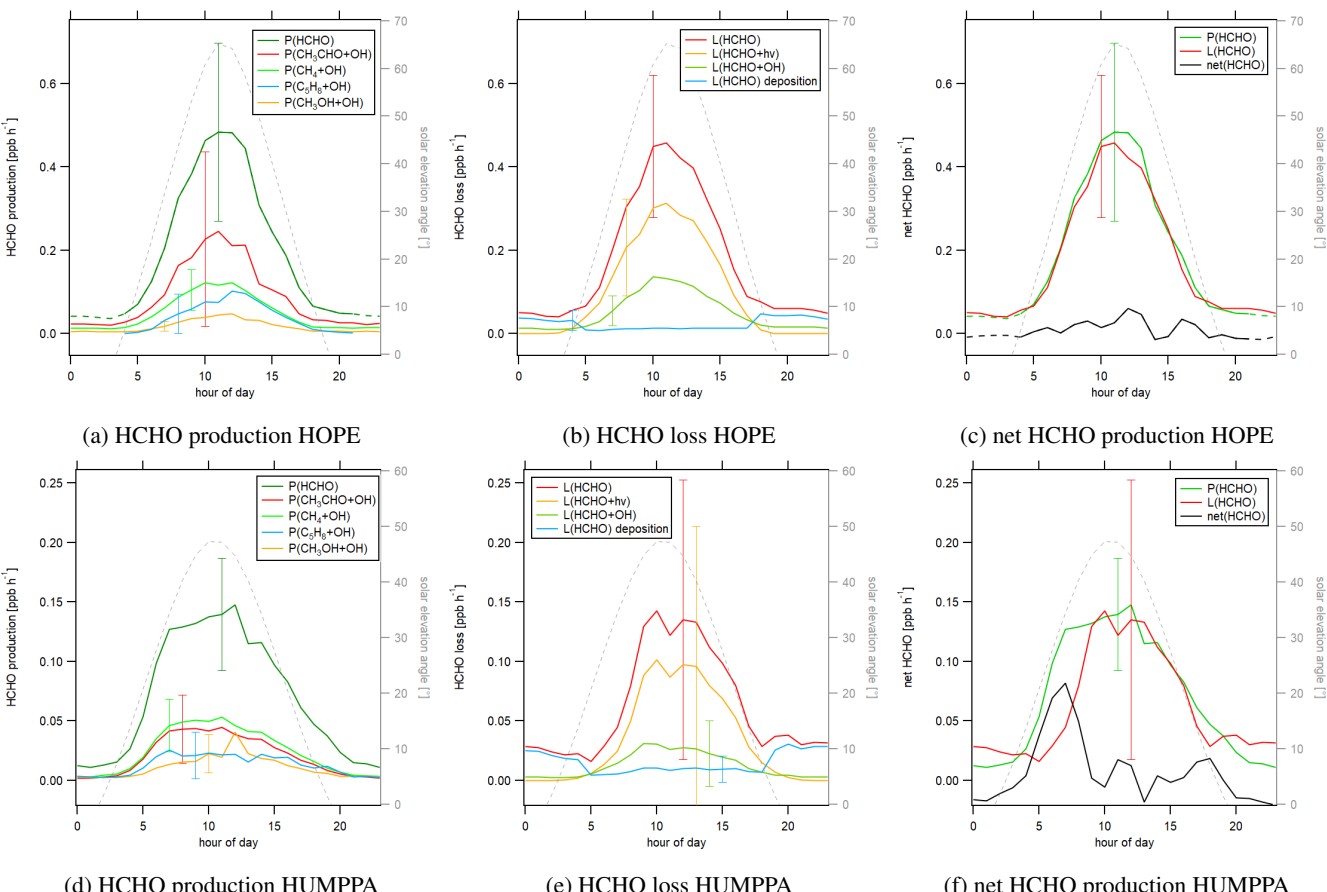

**Figure 9.** Diurnal profiles of HCHO production P(HCHO) and loss L(HCHO) terms during the HOPE campaign 2012 and the HUMPPA campaign 2010.

ratio during the CYPHEX campaign, but the variability was higher which is likely due to the longer time period of available data which was about four weeks for HOPE (compared to a bit more than one week for CYPHEX) including e.g. larger temperature variations. HCHO mixing ratios in Hyytiälä during the HUMPPA campaign ranged between 0.03 and 5.7 ppb$_v$ with an average of $0.4 \pm 0.5$ ppb$_v$. The variability is high because of biomass burning events in Russia which were detected at the

site in Finland on July 26 - 30 and August 9 as described by Williams et al. (2011). The large HCHO peaks detected on these days can also be seen in Figure S1c of the Supplement. Figure S2b and S2c of the Supplement present the diurnal cycles for HCHO including the rate of change dHCHO/dt during HOPE and HUMPPA. The uncertainty is dominated by the atmospheric variability represented by the $1\,\sigma$ standard deviation which is on average 44 % for HCHO for HOPE and 106 % for HUMPPA.

Figure 9 shows the HCHO production and loss rates for the two campaigns. For all cases, the uncertainty was dominated

by the atmospheric variability ($1\,\sigma$) which was 42 % for daytime HCHO production and 43 % for daytime HCHO loss in Hohenpeißenberg and 38 % and 78 % in Hyytiälä, respectively. The measurement uncertainty was around 30 to 40 %. We





present the atmospheric variability by the error bars in Figure 9. For better clarity, we only show one error bar for each term. An overview of all calculated uncertainties can be found in Table S3 of the Supplement. HCHO production during HOPE is shown in Figure 9a. The maximum production rate of $0.48\,\mathrm{ppb}_v\,\mathrm{h}^{-1}$ was reached between 11:00 and 12:00 UTC and comparison to

$j(\mathrm{NO}_2)$ shows good agreement with local noon. The production of HCHO was dominated by the oxidation of acetaldeyhde which contributed to peak production by around 50 %, followed by methane, isoprene and methanol. HCHO loss is shown in Figure 9b. The maximum loss rate of HCHO was $0.46\,\mathrm{ppb}_v\,\mathrm{h}^{-1}$. During the day, HCHO loss was dominated by photolysis and oxidation while at nighttime, deposition was the main loss path for formaldehyde in Hohenpeißenberg. Figure 9c shows HCHO net production during HOPE. The calculated production of HCHO slightly prevailed over its loss. At 11:00 UTC,

HCHO loss was around 95 % of HCHO production. Overall absolute loss and production terms were very similar compared to the results obtained for the site in Cyprus. The main difference is the composition of the HCHO production which was dominated by the oxidation of acetaldehyde in Hohenpeißenberg and by the oxidation of methane in Cyprus. HCHO production and loss during the HUMPPA campaign are shown in Figure 9d - 9f. HCHO production reached a peak value of $0.15\,\mathrm{ppb}_v\,\mathrm{h}^{-1}$ at 12:00 UTC. Methane and acetaldeyhde contributed to the overall production by approximately equal parts followed by

isoprene and methanol. Axinte (2016) showed that the contribution from terpene oxidation to HCHO production was small (1 - 2 % each) which is in line with our findings for the CYPHEX campaign. HCHO loss was dominated by photolysis during the day and by dry deposition at night. Figure 9f shows that HCHO production and loss were in good agreement throughout the day ($\sim$ 90 %) apart for the morning hours 06:00 - 07:00 UTC when HCHO production prevailed over its loss by around 2 to 3 times indicating a missing "loss" term, most likely due to dilution with HCHO-poor air during the rise of the planetary boundary layer

in the early morning hours (accompanied by a peak in the HCHO rate of change as shown in Figure S2b of the Supplement). Overall production and loss terms for HCHO were around 3 times smaller compared to the values obtained for the sites in Germany and Cyprus. We also observed smaller concentrations of $CH_4$, OH radicals and HCHO. It is notable that HCHO production was slightly higher than the loss for both the HUMPPA and the HOPE campaign. This is also reflected by peaks in the rate of change ($0.1 - 0.2\,\mathrm{ppb}_v\,\mathrm{h}^{-1}$) in the morning/midday hours (Figure S2b and c of the Supplement) and could indicate

a transport effect from areas with lower HCHO concentration, e.g. entrainment, according to Equation (6). For HUMPPA, this idea is supported when excluding the data impacted by biomass burning which we show in Figure S7a of the Supplement. It can be seen that HCHO production prevailed over its loss suggesting a missing loss term. The difference was highest in the morning hours with approximately $0.1\,\mathrm{ppb}_v\,\mathrm{h}^{-1}$ and decreased throughout the day indicating e.g. a vertical dilution from higher, HCHO-poor altitudes. For comparison, Figure S7b shows the HCHO production and loss terms when only considering

data impacted by biomass burning. Due to high HCHO mixing ratios the loss terms were substantially higher compared to its production. In periods influenced by biomass burning, the highest HCHO yield was from acetaldeyhde (averaged diel mixing ratios of $1.1\,\mathrm{ppb}_v$ compared to $0.6\,\mathrm{ppb}_v$ without biomass burning); when excluding biomass burning, the yield from methane was slightly higher.





Figure 10. Diurnal profiles of O$_3$ production P(O$_3$) and loss L(O$_3$) terms during the HOPE campaign 2012 and the HUMPPA campaign 2010.

### 3.3.2 Net O$_3$ production

Ozone concentrations in Hohenpeißenberg were on average $44 \pm 11$ ppb$_v$ with a minimum of 5 and a maximum of 97 ppb$_v$. The campaign averaged, diel profile displayed a peak of 48 ppb$_v$ at 16:00 UTC and a minimum of 39 ppb$_v$ at 7:00 UTC. In Hyytiälä, ozone concentrations were between 20 and 76 ppb$_v$ and $42 \pm 11$ ppb$_v$ on average. A diel peak of 49 ppb$_v$ was reached between 15:00 and 16:00 UTC and a minimum of 34 ppb$_v$ at 5:00 UTC. The temporal development of ozone concentrations during the campaigns and the diurnal averages can be found in Figure S3 and S4 of the Supplement. The uncertainty is dominated by the

atmospheric variability ($1\sigma$) which is on average 24 % for O$_3$ for HOPE and 23 % for HUMPPA.

Figure 10 shows ozone production and loss terms for the research campaigns HOPE and HUMPPA. The overall uncertainty was again dominated by the atmospheric variability ($1\sigma$) which was 58 % for daytime O$_3$ production and 83 % for daytime O$_3$ loss in Hohenpeißenberg and 44 % and 51 %, respectively, in Hyytiälä. In comparison, the measurement uncertainty regarding overall production and loss was between 10 and 15 %. A detailed overview of the uncertainties can be found in Table S3 of the





Supplement. We adopted the dry deposition velocity for ozone in Hyytiälä from Rannik et al. (2012). No literature values for ozone dry deposition in Hohenpeißenberg were available. Therefore, we applied the same values as for the HUMPPA campaign which increased the uncertainty of the analysis. However, the fraction of ozone dry deposition of the overall loss was small. For the applied deposition velocity, dry deposition contributed by around 4 % during daytime. For a change in $v_d$ of $\pm\,100\,\%$, the fraction varied between 2 and 7 %. We therefore assume a maximum additional uncertainty of 5 % resulting from this estimate.

Ozone production in Hohenpeißenberg reached peak values of $38\,\mathrm{ppb}_v\,\mathrm{h}^{-1}$. In contrast, ozone loss showed a maximum of only $21\,\mathrm{ppb}_v\,\mathrm{h}^{-1}$ most of which was due to the reaction with NO. In Hyytiälä, ozone production was $6.7\,\mathrm{ppb}_v\,\mathrm{h}^{-1}$ at its diurnal maximum while ozone loss reached $5.1\,\mathrm{ppb}_v\,\mathrm{h}^{-1}$ and was mainly composed of the loss via NO, followed by dry deposition. The differences between production and loss terms could again indicate a transport effect according to Equation (10) as described for HCHO. The rate of change for $O_3$ showed peak values ($2\text{-}3\,\mathrm{ppb}_v\,\mathrm{h}^{-1}$) during morning/midday hours which was not observed

for the CYPHEX campaign (Figure S4 of the Supplement). $O_3$ production and loss terms during HUMPPA and CYPHEX were similar while the values were significantly higher during HOPE. $O_3$ production in Hohenpeißenberg was almost one order of magnitude higher compared to the other sites which was likely due to the higher ambient $NO_x$ concentrations. The net $O_3$ production (the difference between $O_3$ production and loss) at each site could give a hint regarding the dominant chemical ozone regime. For HOPE, net $O_3$ production was significantly above zero with diurnal maximum values of around $20\,\mathrm{ppb}\,\mathrm{h}^{-1}$

at 10:00 UTC which could indicate a VOC limitation. In contrast, net $O_3$ production was close to zero for CYPHEX and the ozone regime was more likely $NO_x$ limited. We will discuss the dominant chemical ozone regime in detail in the following section.

### 3.4 Chemical regime

Various methods exist for determining the prevailing chemical ozone regime (i.e. the net efficiency of production or loss),

one of which is the ratio of formaldehyde and nitrogen dioxide. We have calculated the diel HCHO/$NO_2$ ratios for all three measurement sites which is shown in Figure 11a. The HCHO/$NO_2$ ratio was highest with $8.0 \pm 2.4$ in Cyprus, followed by $1.4 \pm 0.7$ in Hyytiälä and $0.7 \pm 0.2$ in Hohenpeißenberg. According to findings by Martin et al. (2004) and Duncan et al. (2010) these values indicate a dominant $NO_x$ limited regime during CYPHEX and a dominant VOC limited regime during HOPE. The regime during HUMPPA was likely transitioning between both limitations. We show two additional approaches

for determining the present chemical regime in Figure 11. In Figure 11b, we present the daytime averages of $\alpha_{CH_3O_2}$ versus the NO concentration. All campaigns show a linear correlation. Daytime average NO concentrations during CYPHEX ranged from 1 to $45\,\mathrm{ppt}_v$ accompanied by an increase in $\alpha_{CH_3O_2}$ from around 55 to 75 %, giving a slope of $5.54\,\mathrm{ppb}_v^{-1}$. The slope for the increase in $\alpha_{CH_3O_2}$ with NO is approximately half for the HUMPPA campaign with a value of $2.52\,\mathrm{ppb}_v^{-1}$. The NO concentration ranged from 22 to $76\,\mathrm{ppt}_v$ along with an increase in $\alpha_{CH_3O_2}$ from 73 to 87 %. Finally for the HOPE campaign,

NO concentrations and its range were highest with values between 60 and $350\,\mathrm{ppt}_v$. At the same time, $\alpha_{CH_3O_2}$ showed the smallest increase by only 3 percentage points. The resulting slope was smallest with a value of $0.11\,\mathrm{ppb}_v^{-1}$. $\alpha_{CH_3O_2}$ indicates the share of $CH_3O_2$ that forms HCHO, predominantly through the reaction with NO. The competing reaction is the conversion with $HO_2$ to $CH_3OOH$. A high value for $\alpha_{CH_3O_2}$ which is not or only little responsive to a changing NO concentration indicates





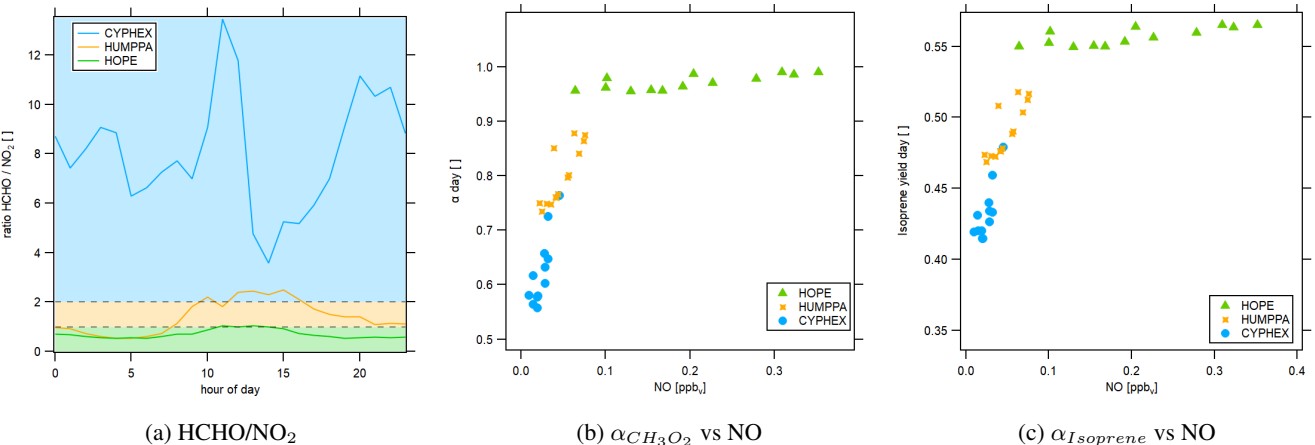

(a) HCHO/$NO_2$      (b) $\alpha_{CH_3O_2}$ vs NO      (c) $\alpha_{Isoprene}$ vs NO

**Figure 11.** Determination of the dominant chemical regime via (a) the HCHO/$NO_2$ ratio (colored areas indicate the dominant chemical regime according to Duncan et al. (2010) which are blue (HCHO/$NO_2 > 2$) for a $NO_x$ limitation, green (HCHO/$NO_2 < 1$) for a VOC limitation and yellow ($1 < $ HCHO/$NO_2 < 2$) for the transition), (b) the fraction of methyl peroxy radicals forming HCHO and (c) the HCHO yield from isoprene.

a VOC limited regime while in a $NO_x$ limited regime, small changes in ambient NO have a large effect on the HCHO formation from $CH_3O_2$. In analogy, we show the HCHO yield from isoprene according to Equation (3) versus NO concentrations in Figure 11c which we suggest as an indicator for the present chemical regime, too. For the CYPHEX campaign, the isoprene yield was most responsive to ambient NO concentrations with a slope of $1.61\,ppb_v^{-1}$ indicating a $NO_x$ limitation. In contrast, the isoprene yield during the HOPE campaign was almost non-responsive to changing NO indicating a VOC limitation (slope of $0.05\,ppb_v^{-1}$). These methods to determine the dominant chemical regime only require the knowledge of a small number of trace gas concentrations and parameters, which are NO, OH, $HO_2$ and the ambient temperature.

## 4 Conclusions

In this study, we have analyzed the photochemical processes contributing to formaldehyde and ozone production and loss across Europe based on in-situ trace gas observations during three different stationary field campaigns in Cyprus (CYPHEX 2014), Germany (HOPE 2012) and Finland (HUMPPA 2010). Very consistently across all sites, we found that formaldehyde loss can be well balanced by the production via OH oxidation of methane, acetaldehyde, isoprene and methanol. Formaldehyde loss is represented by photolysis, OH oxidation and to a small part by dry deposition. Ozone chemistry is mainly controlled by nitrogen oxides. The production can be described by $NO_2$ photolysis and the loss is mainly a function of NO reduction and to a smaller part of photolysis and dry deposition. We found a good agreement between $O_3$ production and loss in Cyprus and Finland, while the production was approximately double its loss in Southern Germany. Finally, we have presented several different approaches for determining the prevalent chemical regime which included the HCHO/$NO_2$ ratio as well as the fraction of $CH_3O_2$ forming HCHO and the HCHO yield from isoprene in dependence on the ambient NO concentration. We identify





a VOC limited regime during the HOPE campaign in Germany and a $NO_x$ limited regime during the CYPHEX campaign in Cyprus, whereas chemistry during the HUMPPA campaign in Finland was likely at a transition point.

While ongoing research on HCHO photochemical processes has continuously widened the contributors to possible production paths and the complexity of calculations and models, we show that the consideration of only four precursor VOC is capable of almost completely representing the HCHO production term at various sites across Europe. We encourage to widen HCHO budget calculations based on in-situ trace gas observations to more sites worldwide as a simple, but effective tool to monitor photochemical processes and air quality, including the dominant chemical regime.

*Data availability.* Data measured during the research campaigns CYPHEX, HOPE and HUMPPA are available to all scientists and will be uploaded to https://keeper.mpdl.mpg.de/.

*Author contributions.* HF had the idea. CMN and HF designed the study. CMN analyzed the data and wrote the manuscript. JS, JNC and BB provided J-values. CPD provided trace-gas measurements for HOPE. JNC provided OH and $HO_2$ model data for HUMPPA. JW provided VOC data. HH, CE, AN, KT, MM, CM and LT provided OH and $HO_2$ data for CYPHEX and HOPE. HCHO data were measured and provided by AR and RA. SH provided $CH_3OOH$ data. $NO_x$ and $O_3$ data were obtained from HF.

*Competing interests.* The authors have no competing interests to declare.

*Acknowledgements.* We acknowledge all researchers and supporting personnel who participated in the CYPHEX campaign 2014, the HOPE campaign 2012 and the HUMPPA campaign 2010. We would like to thank Pekka Rantala (University of Helsinki) for providing the $CH_3CHO$ data for HUMPPA.

*Financial support.* The article processing charges for this open-access publication were covered by the Max Planck Society.





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
