# Peer review of "Measurement report: Photochemical production and loss rates of formaldehyde and ozone across Europe"

_Atmospheric Chemistry and Physics, 2021_

## Author Comment (AC1)

**Referee 1**

We thank Referee 1 for taking the time to review our manuscript and the valuable feedback. We have corrected our manuscript according to the referee's comments and think it is now significantly improved.

Nussbaumer et al use observations from three campaigns across Europe (CYPHEX in Cyprus, HOPE in southern Germany, and HUMPAA in Finland) to calculate production of HCHO and O3. They find that, across all locations, HCHO production can be closely approximated with only production from methane, acetaldehyde, isoprene, and methanol. They also find that the ozone production regime varies by location. This is an interesting paper that is suitable for publication in ACP once the minor issues below are addressed.

We would like to thank the referee for the positive feedback and the recommendation for publication.

**Line 37**: You could also cite Fried et al, 2011 who found similar results about methane being the dominant contributor to HCHO production in the remote atmosphere.

Fried, A., et al. (2011). "Detailed comparisons of airborne formaldehyde measurements with box models during the 2006 INTEX-B and MILAGRO campaigns: potential evidence for significant impacts of unmeasured and multi-generation volatile organic carbon compounds." Atmospheric Chemistry and Physics **11**(22): 11867-11894.

We have added this reference.

Lines 37 ff.: Fried et al. (2011) identified methane to be the main precursor of HCHO in remote regions based on model simulations and measurements during the campaign INTEX-B (Intercontinental Transport Experiment-Phase B) in 2006.

**Line 108**: I agree that, since the isoprene yield is bounded, this estimate doesn't have a large impact on the overall message of the paper, but for the case of CYPHEX, this range still gives a factor of 2 difference in the potential values of HCHO production from isoprene. I think it's appropriate to either add more discussion to justify your assumption that [HO2] = [RO2] (box modeling studies from similar environments?) or use the data you have to try and estimate the ratio. An alternative method of calculating $P(O_3)$ is from $k_{NO+HO2}$[NO][HO$_2$] + $k_{NO+RO2}$[NO][RO$_2$]. If you equate production from this method with that from the JNO2 method, you could potentially estimate RO2 that way. For HUMPPA, at least, there also appears to be box modeling results by one of the co-authors that could be used to evaluate this assumption.

Thank you for this suggestion. We have calculated [RO2] from equating P(O3) via NO2 photolysis and via the reaction of NO with HO2 and RO2. We used $k_{NO+CH3O2}$ as estimate for $k_{NO+RO2}$. We have also calculated P(O3) from $k_{NO+HO2}$[NO][HO2] + $k_{NO+RO2}$[NO][HO2]. The resulting production term for O3 shows close agreement to the production calculated from the photolytic reaction of NO2 which provides additional justification for our assumption that [HO2] = [RO2].

[Figure]

We have added these Figures to the Supplement and refer to them in the main text. We have also added text referring to the box model study performed by Crowley et al. (2018) showing that [HO2] ≈ [RO2] during HUMPPA.

Lines 111 ff.: This assumption is justified when looking at O3 production (P(O3)) terms. P(O3) can either be calculated via the photolytic reaction of NO2 as presented in Section 2.2 or via the reaction of HO2 or RO2 with NO. We equate the two terms, using the rate constant of the reaction of NO and CH3O2 as estimate for the reaction of NO and RO2, and calculate RO2. We show the diurnal profiles of HO2 and calculated RO2 in Figure S1a of the Supplement. Conversely, we calculate P(O3) for both cases, equating RO2 to HO2 which show close agreement and can be seen in Figure S1b of the Supplement. This is also confirmed by findings from Crowley et al. (2018) (presented in Figure 9) based on model simulations of HO2 and RO2 during HUMPPA.

**Figure 1:** The light grey labels on the blue background are very difficult to read when you print this out. I would recommend changing the font color.

We agree with the referee and have changed the color to a darker gray.

**Section 2.3.1:** What time resolution are you using for your P(HCHO) and P(O3) calculations? Do you average everything to the photolysis measurement frequency (10 mins?) or something else?

Thank you for pointing this out. We use a 4-minutes time resolution for calculating the production and loss terms which is the integration time of the OH data. All other species used were interpolated to this time stamp. We have added text for clarification and have corrected this in Table S3 of the Supplement.

Lines 212 f.: For the point-by-point calculations, the data were interpolated to the OH time stamp with a 4-minutes time resolution.

**Line 202:** There's an extra ß in Hohenpeißenberg.

Thank you, we have corrected that.

**Line 211:** You need more discussion about how you arrived at the boundary layer heights. At the very least, summarize what was outlined in Fischer et al.

Fischer et al. presents data from five different measurement sites whereas the BLH was measured at four of those sites including CYPHEX and HUMPPA. Unfortunately, there are no BLH measurements for the HOPE campaign available which were estimated based on the observations from the other sites.

Lines 225 f.: These values were derived from BLH measurements at other locations during summertime including the CYPHEX and the HUMPPA measurement site, and a site in central Germany situated at a comparable altitude.

**Line 234:** Similar to the previous comment, how do you come up with the 20% uncertainty for the boundary layer height, if you don't have observations?

The stated uncertainty of 20% presents the cross section of measurement uncertainty and atmospheric variability based on the BLH in situ measurements performed at the four sites discussed in Fischer et al. Since the BLH for HOPE was derived from these measurements, the uncertainty was also derived in this way. However, an estimated value probably does have a higher uncertainty compared to the value it was derived from. We have therefore increased the uncertainty to 30% for the HOPE BLH values.

Line 226 f.: We assume that this estimate increases the uncertainty from 20% (BLH measurements) to 30%.

**Line 272:** Here and elsewhere, when discussing diurnal profiles, it's much more intuitive to use local time instead of UTC, especially since you are referencing multiple sites that have different UTC offsets.

The referee is correct that the sites have different UTC offsets. However, we would like to continue using UTC for a consistent presentation and in order to avoid confusion, for example regarding local summer and winter time. We have defined the mean local time difference in Section 2.3 and in the caption of Figure 3 and have added a reference to it when first discussing diurnal profiles. We also show the solar elevation angle in the diurnal profiles.

Lines 288 f.: Please note the mean local time difference stated in Section 2.3 and Figure 3.

**Figure 5:** Is this in local or UTC? Please label. Also, for panel a, there is a lot of overplotting. Could you redo the figure where production rates from the individual species are stacked on top of each other to prevent the over plotting?

We present all times in UTC and have added a label for clarification. We have added a Figure to the Supplement to show all production terms in individual panels and refer to it in the main text.

Figure 5: Temporal development of (a) HCHO production terms, (b) HCHO loss terms and (c) net HCHO from July 22 to July 31, 2014 during the research campaign

CYPHEX in Cyprus. The NO2 photolysis frequency j(NO2) is shown in (d) as illustration of the diurnal cycle. All times are in UTC.

Lines 295: The individual production terms are shown in Figures S5 of the Supplement.

**Figure 6:** I assume this is for all data averaged together?  Does this include both daytime and nighttime values?  Please indicate.

Yes, the pie chart presents an average of all data, including day- and nighttime values. We have clarified this in the text and the Figure caption.

Lines 307 f.: This is additionally illustrated in Figure 6 which presents the daily average share (including day- and nighttime values) of each production term based on a balance to the overall loss rate.

Figure 6: Chemical production terms of HCHO during CYPHEX including daily averages of all data.

**Line 302**: Where do you get the 20% value from to vary the yields?

While the share of methyl peroxy radicals forming formaldehyde can theoretically be between 0 and 100%, the availability of NO shifts the value towards the upper end. We therefore decided not to include these threshold values in the sensitivity study, but chose a 20% change as an example to demonstrate the sensitivity of the overall result to the $\alpha_{CH3O2}$. We clarified this in the text.

Lines 319 ff.: As an example, a 20% decrease in $\alpha_{CH3O2}$ would give a HCHO production from $CH_3O_2$ of 38% … .

**Line 311:** You attribute the nighttime increase in HCHO to local emissions from traffic.  Looking at the supplementary figure, it looks like on the night of July 27[th], the HCHO concentration nearly doubles.  Even with a low boundary layer, that seems too large to be just mobile emissions.  Have you looked at $\Delta HCHO/\Delta CO$ or $\Delta HCHO/\Delta NOx$ to see if this is in line with what you would expect from traffic emissions?  Couldn't this also just be a new air mass moving in or advection from a different region?  Do you have meteorological observations you could use to look at this?

The referee is correct that we cannot be certain that this increase is attributed to vehicle emissions and we have changed the text accordingly.

Line 328: The reason for this nighttime HCHO increase is not yet fully understood.

Regarding meteorological aspects, we have run the NOAA Hysplit Model for backward trajectories exemplary for July 27 which shows that air originated from above the Mediterranean Sea which does not reveal any particular HCHO nighttime source. Due to the short atmospheric lifetime of HCHO it is also unlikely that primary HCHO emissions from distant locations are transported to the measurement site.

[Figure]

NOAA HYSPLIT MODEL
Backward trajectory ending at 0600 UTC 27 Jul 14
GDAS Meteorological Data

Job ID: 189965         Job Start: Wed Oct  6 12:07:02 UTC 2021
Source 1    lat.: 34.960000   lon.: 32.380000   height: 650 m AMSL

Trajectory Direction: Backward      Duration: 24 hrs
Vertical Motion Calculation Method:       Model Vertical Velocity
Meteorology: 0000Z 22 Jul 2014 - GDAS1

**Line 315:** I think you need a little more discussion about the deposition.  Instead of there being no deposition, couldn't it just be that the deposition rate is lower than the rate of increase from whatever processes is leading to the increased nighttime HCHO, whether it's advection or direct emission?

We agree with the referee that deposition could be counteracted by a nighttime HCHO source such as advection or secondary formation from terpenes via ozone. The HCHO yields from terpenes are not well known and vary greatly in literature, but are likely very low. The value for the deposition can usually be seen as a lower estimate (previously described by Crowley et al, 2018). We have added some text to the manuscript.

Lines 331 ff.: Please note that the effect of deposition processes could also be counteracted by a nighttime HCHO source, such as terpene oxidation by ozone or advection. The determined value for the deposition can therefore be seen as a lower estimate. As we do not observe net loss of HCHO at night during the CYPHEX campaign, we estimate dry deposition to be negligible.

**Line 402**: Do you have thoughts as to why the contribution from acetaldehyde photolysis was so much greater at HOPE vs CYPHEX?  Is it from larger acetaldehyde concentrations or just a result of higher photolysis related to the higher altitude?

Please note that the contribution from acetaldehyde regarding HCHO production is related to oxidation by OH radicals whereas the contribution from acetaldehyde photolysis is negligibly small. The contribution from acetaldehyde oxidation is greater for HOPE due to a higher acetaldehyde concentration which was 1.21 ± 0.64 ppbv throughout the campaign. In comparison, the average acetaldehyde concentration during CYPHEX was 0.38 ± 0.16 ppbv. This is counteracted by higher temperatures and higher OH radical concentrations during CYPHEX. However, the effect of higher acetaldehyde prevailed.

**Line 457:** Duncan et al use ratios of tropospheric column HCHO/NO$_2$ to estimate the ozone production regime. While they do show that this is similar for PBL values, here you are just using surface observations. Please give more justification on how the values you use to estimate the ozone production regime could be affected by this difference. Do you have any evidence that the PBL is well-mixed enough to make this assumption? Also, as you cite in your paper, Schroeder et al have pointed out that the values cited in Duncan vary by location. Do your results change at all when taking this into consideration?

We would like to point out that tropospheric columns of HCHO/NO2 from satellite measurements are usually taken into consideration in the absence of ground-based observations. The dominant ozone regime is related to local chemistry taking place and can change with altitude or geographical location. In-situ observations are more reliable when determining the HCHO/NO2 ratio and deducing O3 sensitivity from it, which was also pointed out by Schroeder et al. The considered thresholds were determined via model simulations which assume a well-mixed boundary layer which makes the use of surface observations beneficial. Schroeder et al. present thresholds variations depending on the location particularly in regard to the transition range between a NOx and a VOC limitation, e.g. they find a threshold range from 1.3 to 4.3 in Houston in contrast to the range 1 to 2 presented by Duncan et al. Our results are not affected by higher threshold ranges as the value for the CYPHEX campaign is much higher with an average of 8 and much lower for the HOPE campaign with an average of 0.7. Further, we present two new parameters for determining O3 sensitivity which confirm the findings from in-situ HCHO/NO2 ratios.

---

## Author Comment (AC2)

**Referee 2:**

We thank Referee 2 for taking the time to review our manuscript and the valuable feedback. We have corrected the manuscript according to the referee's comments.

The manuscript by Nussbaumer et al. provides the first HCHO budget calculations across Europe using *in situ* measurements as opposed to evaluating the HCHO budget using model simulations. Additionally, the authors show that HCHO production is dominated primarily by the oxidation of methane, methanol, acetaldehyde, and isoprene during three campaigns (CYPHEX, HOPE, and HUMPPA) representing a coastal, mountain, and forested site, respectively. The HCHO yield from isoprene and fraction of methyl peroxy radicals (CH3O2) forming HCHO were also shown as an alternative method for determining whether a location is NOx or VOC-limited (or in some transition regime).

General comments: While the oxidation of methane, methanol, acetaldehyde, and isoprene are the dominant VOC precursors to HCHO production, care should be taken throughout the manuscript to never imply that only these four chemical species make up the entirety of the HCHO budget. HCHO is commonly used as a VOC tracer and comes from more than simply those four species as the authors showed with the pie chart in Figure 6. As an example, Line 75 is misleading since it states that "HCHO production can be accounted for by the oxidation of methane, methanol, acetaldehyde, and isoprene" which implies 100%. Rather, it should read "...predominantly accounted for..." or some other conditional phrasing.

We agree with the referee and have clarified that the four mentioned VOC precursors do not account for 100%, but rather dominate the HCHO production.

Lines 76 ff.: We are first to present HCHO budget calculations from in-situ measurements across Europe and show that in all three locations HCHO production can be predominantly accounted for by the oxidation of methane, methanol, acetaldehyde and isoprene.

Lines 363 f.: (…) leads to the conclusion that HCHO production can be approximated by OH oxidation of methane, acetaldehyde, isoprene and methanol.

Lines 500 ff.: Very consistently across all sites, we found that formaldehyde loss can be predominantly accounted for by the production via OH oxidation of methane, acetaldehyde, isoprene and methanol.

The manuscript fits well within the scope of ACP and provides a good, detailed analysis of uncertainty. I recommend publication after attention to the previous general comment and the following specific comments/technical corrections.

We thank the referee for this positive feedback and the recommendation for publication.

Specific Comments:

- For the reader, explicitly define somewhere in the text what is meant by "atmospheric variability" (i.e., what factors control this uncertainty)

We have added text to explain the term "atmospheric variability".

 Besides the uncertainty resulting from the calculation, an additional uncertainty arises from the atmospheric variability which describes ambient, instrumentally independent variations of the considered trace gases and parameters caused by for example atmospheric turbulence.

- Whenever mentioning the detection limit of an instrument (for instance, in Section 2.3.1), the integration time necessary to achieve that detection limit should be mentioned.

Unless otherwise stated, the detection limits refer to the time resolution which is shown in Table S3 of the Supplement. We have added text for clarification.

 All stated detection limits refer to the time resolution shown in Table S3 of the Supplement.

 $CH_3OH$ was measured via ColdTrap PTR-MS with a detection limit of around 50 pptv (integration time of 5 mins).

- Figure 1: There is space above (or below) the HCHO precursors to write the actual chemical name for each of the chemical formulas (i.e., acetone, MHP, etc.)

We have changed the Figure according to the referee's suggestion.

[Figure]

- Line 197: The determination of the acetaldehyde and formaldehyde photolysis frequencies are from a parameterization using IUPAC quantum yield data and measurements of j(NO2) and j(O1D). This parameterization should be explicitly shown in the SI for the reader.

We now present the details on the parameterization in Table S2 and Equation (S1) of the Supplement. We additionally show an example for the performance of the

parameterization in Figure S2 of the Supplement. We have added text in the manuscript for clarification.

Lines 204 ff.: The photolysis frequencies for acetaldehyde j(CH3CHO) and formaldehyde j(HCHO) were determined via parameterizations based on j(NO2) and j(O1D) according to Equation (S1) with the coefficients presented in Table S2 of the Supplement. The latter were derived from least-squares fits to photolysis frequencies from a large set of spectroradiometer measurements at Jülich, Germany (Bohn et al., 2008) under all weather conditions and were originally derived for the HUMPPA campaign. In this work more recent quantum yields for the HCHO photolysis as recommended by IUPAC (2013) were used with an estimated uncertainty of 20%. An example for the performance of the parameterization is shown in Figure S2 of the Supplement.

Figure S2: Correlations of measured j(HCHO) (molecular plus radical channel) with (a) j(O1D) and (b) j(NO2) from spectroradiometer measurements. The correlation of measured and parameterized j(HCHO) according to Equation (S1) is shown in panel (c). Only one out of ten data points from the original data set is shown for clarity.

[Figure]

Table S2: Coefficients for the calculation of j(HCHO) and j(CH3CHO) according to Equation (S1).

| species | $a_1$ | $a_2$ [s] | $b_1$ | $b_2$ [s] |
|---|---|---|---|---|
| HCHO | 1.719 | $-1.768 \times 10^4$ | $4.701 \times 10^{-3}$ | $-3.471 \times 10^{-2}$ |
| CH₃CHO | $1.516 \times 10^{-1}$ | $-8.970 \times 10^2$ | $4.567 \times 10^{-5}$ | $1.711 \times 10^{-3}$ |

Equation (S1):

$$j(species)_{parameter} = a_1 \times j(O^1D) + a_2 \times (j(O^1D))^2 + b_1 \times j(NO_2) + b_2 \times (j(NO_2))^2$$

- Figure 4: At least for the example shown, it is odd that points were selected when the HCHO mixing ratio was still increasing and were included in the fit. In this

particular case, the slope would be underestimated and the deposition velocity would be biased. What motivated the decision to always select points between 21:00 - 01:30 UTC as opposed to looking at the underlying nighttime HCHO mixing ratio data over several hours?

We intended to find a reproducible nighttime period with decreasing HCHO concentrations which could be applied to all nights. Choosing the data between 21:00 and 01:30 UTC for HOPE and between 00:00 and 04:30 UTC for HUMPPA is the best compromise regarding all considered nights. However, when individually choosing the time period for each night, the deposition velocity for HOPE rises to 0.94cm-1 during the day and 0.47cm-1 during the night (compared to 0.40 cm-1 and 0.20 cm-1 for a constant time period). For HUMPPA, we get 0.87cm-1 during the day and 0.43cm-1 during the night (compared to 0.72 cm-1 and 0.36 cm-1 for a constant time period). We have updated our calculations and the Figures using the new velocities according to the referee's suggestion. There are no significant changes to the results discussed in the manuscript.

[Figure]

Lines 237 f.: (…) the nighttime deposition velocity according to Equation (11) which gives vd(day)=0.94cm s-1 and vd(night)=0.47cm s-1.

Lines 274 f.: The deposition velocity was determined in analogy to the HOPE campaign based on the nighttime HCHO loss on the basis of 14 nights and was 0.85cm s-1 during the day and 0.43cm s-1 during the night.

- Lines 395-396: Please create pie charts for HOPE and HUMPPA (as done for CYPHEX) since this readily shows the contributions of the four dominant precursors as well as the other reactions included in your chemical mechanism. Could either place resulting figure in main text or SI.

The pie chart we have created for CYPHEX is based on a balance to the HCHO loss which worked well because the summed production terms are smaller compared to the loss terms. However, for HUMPPA and HOPE, we find that the HCHO production term approximated by oxidation of the four mentioned VOC precursors is slightly higher compared to the overall loss term. We suggest this effect to be the result of a

transport effect from areas with lower HCHO concentrations. We show the pie charts based on the summed HCHO production terms in Figure S10 of the Supplement and refer to it in the main text.

Lines 414 f.: We show a pie chart representing the contribution of the single HCHO production terms during HOPE in Figure S10a of the Supplement.

Lines 424: Figure S10b shows the share of the individual HCHO production terms during HUMPPA.

[Figure]

Line 475: Explicitly state that specialized instrumentation is still required (particularly for OH and HO2) for these alternative methods of determining the chemical regime.

We have added text to indicate that the measurement of NO, OH and HO2 needs particular instruments.

Lines 494 ff.: Although specialized instrumentation is still necessary to measure NO, OH and HO2, these methods to determine the dominant chemical regime only require the knowledge of a small number of trace gas concentrations and the ambient temperature.

Technical Corrections:

Throughout text: Formatting for d[HCHO]/dt is inconsistent (for example, line 260 and 275)

We have unified the format to $\frac{d[HCHO]}{dt}$.

Line 404: Misspelling of acetaldehyde

Thank you, we have corrected this.

Figure S7A: Please clarify whether the data is from CYPHEX or HUMPPA

Thanks for pointing this out, these are the data for HUMPPA. We have corrected the Figure title accordingly.

Figure S11: Diurnal HCHO production and loss during HUMPPA (…)

Figures: Font size on axes should be increased since the axes are hard to read when printed

We have increased the font size of all axis labels.

Figure S11: Diurnal HCHO production and loss during HUMPPA (…)

Figures: Font size on axes should be increased since the axes are hard to read when printed